# A Multicentre, Prospective, and Retrospective Registry to Characterize the Use, Effectiveness, and Safety of Dalbavancin in German Clinical Practice

**DOI:** 10.3390/antibiotics11050563

**Published:** 2022-04-22

**Authors:** Frank Hanses, Sebastian Dolff, Janina Trauth, Michael Seimetz, Stefan Hagel

**Affiliations:** 1Emergency Department, University Hospital Regensburg, Franz-Josef-Strauß-Allee 11, 93053 Regensburg, Germany; frank.hanses@klinik.uni-regensburg.de; 2Department for Infectious Diseases and Infection Control, University Hospital Regensburg, Franz-Josef-Strauß-Allee 11, 93053 Regensburg, Germany; 3Department of Infectious Diseases, West German Centre of Infectious Diseases, University Hospital Essen, University of Duisburg-Essen, Hufelandstr. 55, 45122 Essen, Germany; sebastian.dolff@uk-essen.de; 4Division of Infectious Diseases, Department of Internal Medicine II, University Hospital Giessen and Marburg, Klinikstraße 33, 35392 Giessen, Germany; janina.trauth@innere.med.uni-giessen.de; 5Advanz Pharma Germany GmbH, Herforder Str. 69, 33602 Bielefeld, Germany; michael.seimetz@advanzpharma.com; 6Institute for Infectious Diseases and Infection Control, Jena University Hospital—Friedrich Schiller University Jena, Am Klinikum 1, 07747 Jena, Germany

**Keywords:** dalbavancin, registry, acute bacterial skin and skin structure infection (ABSSSI), real-world setting

## Abstract

The antibiotic dalbavancin is approved for intravenous treatment of adults with acute bacterial skin and skin structure infections. This study aimed to observe the use, effectiveness, and safety of dalbavancin in clinical practice in Germany. It was a multicentre, prospective, and retrospective registry and consecutively enrolled patients treated with dalbavancin. Each patient was observed from the first to the last dose of dalbavancin, with a 30-day follow-up. Patient inclusion was planned for 2 years, but was terminated early due to low recruitment. All analyses were descriptive. Between November 2018 and December 2019, nine patients were enrolled. Only three patients were treated for the approved indication. Outcome was assessed by the physicians as ‘success’ in five (55.6%) patients, ‘failure’ in one (11.1%) patient, and non-evaluable in three (33.3%) patients. Although the success rate of dalbavancin was lower than reported previously, this may be due to the severity of underlying infections and patients’ high Charlson Comorbidity Index. None of the two reported adverse events were considered related to dalbavancin. These findings were in line with real-world data for dalbavancin from other countries, supporting the drug’s positive benefit–risk profile and suggesting frequent off-label use in German routine practice.

## 1. Introduction

Dalbavancin is a semisynthetic lipoglycopeptide antibiotic that interferes with cell wall synthesis by binding to the D-alanyl-D-alanine terminus of the stem pentapeptide in nascent cell wall peptidoglycans, thereby preventing cross-linking [1]. Pivotal clinical trials demonstrated its efficacy and safety in the treatment of adult patients with acute bacterial skin and skin structure infections (ABSSSI) caused by susceptible isolates of Gram-positive microorganisms [2,3,4]. In February 2015, the European Medicines Agency (EMA) approved dalbavancin for this indication. Due to its long terminal elimination half-life of 14.4 days [1], dalbavancin is recommended to be administered as either a two-dose regimen (1000 mg and 500 mg) with a 1-week interval or a single-dose regimen (1500 mg). Each dalbavancin infusion is administered as a 30 min intravenous (IV) infusion [5].

Dalbavancin’s single- or two-dose regimen may improve the current burden of infusion care experienced by patients, clinicians, and healthcare systems by enabling earlier discharge of hospitalized patients requiring ongoing parenteral antibiotic therapy, outpatient management of patients who would traditionally be admitted for IV antibiotic therapy [6,7], and by avoiding healthcare-associated infections commonly associated with peripheral or central line catheters. Furthermore, health economic benefits may be achieved by shortening the length of stay in the hospital [8,9,10]. These reasons combined with broad activity against Gram-positive bacteria may explain why several studies in various countries have found a high rate of off-label use of dalbavancin [11,12,13,14,15,16,17]. Data on dalbavancin resistance can be found in the following references, among others [18,19].

In Germany, routine clinical use of dalbavancin has not yet been comprehensively described. Therefore, the *Dalbavancin Utilization Registry in Germany* was designed with the aim to collect real-world data on the use of dalbavancin, as well as its effectiveness and safety in routine clinical practice in Germany. The results of this registry—although stopped early due to low patient recruitment—were consistent with real-world safety and effectiveness of data for dalbavancin in other countries; they are in line with the favourable benefit–risk profile of dalbavancin and reflect frequent off-label use in German routine practice.

## 2. Results

### 2.1. Patient Inclusion, Study Duration, and Early Termination

A total of nine patients were recruited between 21 November 2018 and 5 December 2019 at three study sites in Germany. Whereas a 2-year recruitment was initially planned, the *Dalbavancin Utilization Registry in Germany* was terminated early due to low recruitment. The last patient completed the study on 24 July 2020.

Eight patients were retrospectively enrolled (informed consent after the last dalbavancin administration), while one patient gave informed consent after the day of the first dalbavancin administration and was partly observed prospectively. All nine patients of this ‘All Patients Set’ were included in the ‘Full Analysis Set’ and the ‘Safety Analysis Set’, which were identical. No patient discontinued the study prematurely.

### 2.2. Patient Characteristics at Baseline

The median age of patients was 58 years (Table 1). The majority, seven (77.8%) patients, were men. Remarkably, the median Charlson Comorbidity Index Score was 7 (range: 2–9), a relatively high score indicating a high expected mortality rate (Table 1). Three patients had a history of renal failure; two of those were on haemodialysis.

### 2.3. Disease Characteristics

The types of infection treated by dalbavancin were ABSSSI (3/9), endocarditis (3/9), vascular graft infection (2/9), and prosthetic joint infection (1/9) (Table 2, column 2). Concomitant bacteraemia was reported in five (56%) patients. The main Gram-positive pathogens identified were coagulase-negative staphylococci (six patients), Staphylococcus aureus (three patients), and Enterococcus faecium (one patient) (Table 2, column 3).

### 2.4. Antibiotics before, during, and after Dalbavancin Treatment

Before dalbavancin treatment, eight patients received antibiotics for the primary diagnosis (unknown for one patient). Concomitantly with dalbavancin treatment, five patients received other antibiotics (Table 2, column 5). Within 30 days after dalbavancin treatment, six patients received antibiotics (unknown for two patients).

### 2.5. Usage of Dalbavancin

For three patients, one dalbavancin infusion was recorded; for three patients, two infusions; and for three patients, more infusions (for details, see Table 2, column 6).

For seven patients, dalbavancin treatment was a subsequent-line therapy (data missing for one patient). The reason for switching to dalbavancin in the seven patients with subsequent-line therapy was ‘insufficient therapeutic effect of prior therapy’ for four patients and ‘adverse reaction to prior therapy’ for three patients (Table 2, column 7).

### 2.6. Effectiveness of Dalbavancin

For five (55.6%) patients, the physician assessed the treatment outcome as ‘success’ (defined as ‘cured’ or ‘improved’). One (11.1%) patient had the outcome ‘failure’ (defined as ‘worsening’, ‘failure’ or ‘other’). The outcome was non-evaluable in the remaining three (33.3%) patients (Table 2, column 8); one of these patients died from a fatal adverse event, one was hospitalized due to an adverse event, and one was lost to follow-up.

### 2.7. Hospitalization

During dalbavancin treatment, seven (87.5%) patients were hospitalized, while one (12.5%) patient was not hospitalized (data for one patient missing; percentages refer to non-missing patients). The median duration of hospitalization of the seven hospitalized patients was 14 days (range: 2–25). Five (62.5%) patients were in the intensive care unit (ICU) during the study, while three (37.5%) patients were not in the ICU (data for 1 patient missing; percentages refer to non-missing patients). The median duration of ICU stay of the five hospitalized patients was 7 days (range: 2–24).

For one patient, a readmission to hospital within 30 days after end of treatment was recorded. The reason for hospitalization was ‘fever and candida albicans infection’.

### 2.8. Safety

Adverse events were evaluated from first treatment dose up to 30 days after the last dose. Overall, two adverse events were recorded for two (22.2%) patients (Table 3). Both adverse events were serious and judged by the physicians not to be related to dalbavancin. One of the adverse events was fatal, i.e., ‘multiple organ dysfunction syndrome’ in a 60-year-old male with a Charlson Comorbidity Index of 8. The fatal adverse event started 8 days after the third dalbavancin infusion; the patient died the next day. No adverse event led to a change in dalbavancin treatment.

## 3. Discussion

### 3.1. Limitations of the Dalbavancin Utilization Registry in Germany

The study planned to enroll approximately 150 patients during the 2-year recruitment phase, but it was only possible to include 9 eligible patients at three study sites. Due to the relatively recent market launch in Germany (November 2016), at the time of study planning, insufficient data were available for a more precise planning of the sample size. It turned out that the inclusion of study sites and enrollment of patients in Germany was more difficult than expected. Possible reasons may include the generally less frequent use of dalbavancin in Germany, high costs and difficulties in reimbursement, the low rate of infections with multi-drug resistant Gram-positive microorganisms, the traditional tendency towards more frequent and longer hospitalization of patients with chronic diseases, the collection of informed consent in retrospectively enrolled patients, and the reluctance of hospital staff to participate in observational studies.

The *Dalbavancin Utilization Registry in Germany* was terminated early due to low recruitment. The relatively small number of included patients is a major limitation of the study that affects the evaluation of all outcomes. The low number of patients and study sites also enhances potential bias, particularly a selection and information bias. This results in limited generalizability.

The majority of patients (*n* = 8) were included after their last dose of dalbavancin. In these patients, the risk of missing data is high due to probably incomplete recording of adverse events and the lack of possibility of monitoring. Conversely, it is particularly clear in these patients that inclusion in the registry did not influence the treatment decision.

A strength of the study design is that the registry indeed collected real-world data from clinical practice. Although the number of patients is limited, the study provides initial information on dalbavancin application in routine clinical practice in Germany with regard to the study objectives. In order to draw comprehensive conclusions, however, a broader basis of data will be required.

### 3.2. Context of Previous Observational Studies on Dalbavancin

A summary of 10 relevant similar observational studies on the use of dalbavancin in clinical practice in various countries is shown in Table 4.

In the published real-world studies listed in Table 4, the median patient age was between 47 and 76 years and ≥50% of the patients were men, which is in line with the present registry. The median Charlson Comorbidity Index, where analyzed, was 3 and lies markedly below the median Charlson Comorbidity Index of 7 in this study, indicating a particularly high burden of concomitant diseases in the German registry. Like the present study, the previously published studies frequently observed dalbavancin treatment outside the approved indication ABSSSI. Commonly treated indications other than ABSSSI varied between the studies and included device-related infections, endocarditis, and osteomyelitis (Table 4). Overall, it appears that in clinical practice primarily complicated patients and infections are treated with dalbavancin as reflected by age, a high number of comorbidities, and complex underlying infectious diseases.

The observed regimens were diverse and often diverged from the approved dosage for dalbavancin in adults with ABSSSI in the published studies (Table 4). More than two doses were frequently administered, including weekly or bi-weekly doses over several weeks, as also observed in the present study. Previous and concomitant antibiotic treatment was diverse within each study and probably complicates a uniform evaluation of the effectiveness of dalbavancin within studies and the comparability between studies. Success rates of dalbavancin were high in most published studies. The success rate of 55.6% in the present study is markedly lower than in the published studies with [20] as the only exception. The relatively low success rate in the present study may be explained by the low patient numbers, the high Charlson Comorbidity Index, and specific pathologies treated in the present study.

Adverse events were rarely reported in previously published studies and were diverse. Similarly, only two adverse events in two (22.2%) patients with no related adverse events were recorded in the current study. Overall, the low number of adverse events confirms that dalbavancin is well tolerated.

### 3.3. Conclusions

The findings of the *Dalbavancin Utilization Registry in Germany* are in line with the growing body of published real-world safety and effectiveness data for dalbavancin. Despite all limitations (see Section 3.1), the results support the positive benefit–risk balance profile of dalbavancin. The results confirm the predominant treatment of off-label indications and use of off-label dose regimens as a back-up option even in pre-treated and multi-morbid patients previously observed in routine practice in various other countries (see Section 3.2; Table 4).

## 4. Materials and Methods

### 4.1. Study Design and Patients

The *Dalbavancin Utilization Registry in Germany* was a multicentre, prospective, and retrospective observational registry of adult patients treated with dalbavancin in Germany, as prescribed by the physicians according to local clinical practice. The registry was designed to observe the use, safety, and effectiveness of dalbavancin. Prescription of dalbavancin was solely at the discretion of the physicians and no additional procedures or tests were required to be performed by the observational study protocol. Patient data were collected both retrospectively (since launch in Germany in November 2016) and prospectively. The rationale for retrospective data collection was the intended reflection of the current real-world setting and the timely completion of the registry. According to the German Medicines Act, prospective data could only be collected for in-label indications.

This registry enrolled male and female patients who were ≥18 years of age at the time of receipt of dalbavancin, who received at least one infusion of dalbavancin, and who signed the informed consent form. Patients who participated in a clinical trial in which treatment for dalbavancin was managed through a protocol were not eligible for this registry. Both retrospectively and prospectively included patients had to fulfill the same selection criteria. Patients were enrolled consecutively whenever possible. For each patient, the study comprised three periods: baseline (defined as the time the specified primary infectious disease was diagnosed), treatment period (variable time interval from the first to the last dalbavancin dose), and follow-up (defined as the 30-day period following end of intravenous dalbavancin administration).

This registry was planned to be conducted at 10–15 study sites in Germany. Due to their relatively small number, all identified potential study sites using dalbavancin were invited to participate. The total study duration was expected to be approximately 2 years. However, the *Dalbavancin Utilization Registry in Germany* was terminated early when it became clear that the recruitment target would not be met in a reasonable timeframe. At registry closure, seven sites were initiated with three sites having recruited patients.

The study protocol was approved by the Ethics Committee at the Medical Faculty of the University of Rostock, Germany, on 14 February 2018 (registration number: A 2018-0024). The study was first posted at ClinicalTrials.gov (identifier: NCT03696901) on 5 October 2018, before the inclusion of the first patient.

### 4.2. Variables and Data Sources

The variables of this registry included patient characteristics (demographics, height and weight, medical history, Charlson Comorbidity Index), disease characteristics (primary infectious disease diagnosis for which the patient received dalbavancin, the date of diagnosis or time of onset), pathogen characteristics, antibiotic therapy (from diagnosis of the primary infectious disease through 30 days after the last dalbavancin administration), dalbavancin usage (including dose, number of infusions, duration of infusion, administration schedule, reason for use), physician-assessed clinical outcome (categories: cured, improved, failure, non-evaluable, worsening, other), concomitant medication, hospitalization, and adverse events (from first treatment dose up to 30 days after last dose).

The physician-assessed outcome ‘clinical response’ was categorized as success (assessment ‘cured’ or ‘improved’), failure (assessment ‘failure’, ‘worsening’, or ‘other’), or non-evaluable. Missing assessments of clinical outcome for the clinical response analysis were obtained from the physicians post-hoc in writing and used for the post-hoc clinical response analysis.

Generally, all data originated from patient medical records and were entered in electronic case report forms by the physicians or their authorized personnel. All adverse events and medical terms reported in this study were coded using the Medical Dictionary for Regulatory Activities (MedDRA version 21.1). Medications (antibiotic medication and relevant prior and concomitant medications) were coded using the World Health Organization Drug Dictionary Global (B3 format version) 2018.

### 4.3. Statistics

All analyses were carried out descriptively. Analyses were based on observed data without imputation. Missing data were displayed in the tables and listings along with the number of patients with missing data, but missing information was not included in the evaluation of the variable (e.g., percentages or medians). All summarizations and analyses were performed using SAS^®^ Version 9.4.

The originally planned sample size of this registry was approximately 150 patients. This was the estimated number of eligible patients who could be enrolled at 10–15 sites in Germany in a 2-year recruitment phase. All analyses were performed with the safety analysis set, which included all patients who had received at least one dose of study medication and for whom informed consent was available (here referred to as the all patients set).

## Figures and Tables

**Table 1 antibiotics-11-00563-t001:** Demographic characteristics (all patients).

Variable	All Patients (*N* = 9)
Age [years]	
*n* (missing)	9 (0)
Mean (SD)	56.9 (10.6)
Median (range)	58 (33–71)
Age category, *n* (%)	
18–49 years	1 (11.1%)
50–59 years	4 (44.4%)
60–69 years	3 (33.3%)
70–79 years	1 (11.1%)
Sex, *n* (%)	
Male	7 (77.8%)
Female	2 (22.2%)
BMI [kg/m^2^]	
*n* (missing)	7 (2)
Mean (SD)	26.2 (5.1)
Median (range)	25.9 (22.9–30.4)
Charlson Comorbidity Index Score	
*n* (missing)	9 (0)
Mean (SD)	6.6 (3.9)
Median (range)	7.0 (2.0–9.0)

BMI, body mass index; SD, standard deviation.

**Table 2 antibiotics-11-00563-t002:** Per-patient data on patients, disease, therapy, and outcome.

Patient Number	Infectious Disease	Pathogen(s)	Previous Therapy	Concomitant Antibiotics (Combination Partners)	Dalbavancin Regimen (No. of Administrations: Doses in the Order Administered [mg])	Reason for Change to Dalbavancin	Outcome
1	ABSSSI	*Staphylococcus aureus*, *Enterococcus faecium*	Piperacillin/Tazobactam, Cefazolin, Ciprofloxacin, Linezolid, Vancomycin	Vancomycin	2: 1500, 1500 On day 0, 8	Adverse reaction to prior therapy	Non-evaluable
2	ABSSSI	*Staphylococcus epidermidis*	Vancomycin	-	3: 1000, 500, 500 On day 0, 38, 82	Insufficient therapeutic effect of prior therapy	Non-evaluable
3	Surgical site infection (ABSSSI), Gram-positive bacteraemia	*Staphylococcus aureus*, *Klebsiella pneumoniae*, *Proteus vulgaris*, Coagulase-negative staphylococci, *Staphylococcus epidermidis*	Cefazolin, Cefotaxime, Fosfomycin, Rifampicin, Daptomycin, Cefalexin, Meropenem, Linezolid, Piperacillin/Tazobactam	Cefotaxime	2: 1500, 1500 On day 0, 10	Adverse reaction to prior therapy	Success
4	Native valve endocarditis, lung abscess, Gram-positive bacteraemia	*Staphylococcus aureus*	Ampicillin, Meropenem, Cefazolin, Fosfomycin, Flucloxacillin, Ciprofloxacin, Meropenem	-	1: 1500	Insufficient therapeutic effect of prior therapy	Non-evaluable
5	Native valve endocarditis, Gram-positive bacteraemia	*Streptococcus infantarius*	Benzylpenicillin, Sulfamethoxazole/Trimethoprim, Cefazolin	Benzylpenicillin	1: 1500	Unknown	Failure (renewed suspicion of endocarditis)
6	Prosthetic valve endocarditis	Culture negative	Cefazolin, Ampicillin/Sulbactam, Ampicillin	Ampicillin; Sulbactam; Amoxicillin; Clavulanate potassium	2: 1500, 1500 On day 0, 23	Insufficient therapeutic effect of prior therapy	Success
7	Device-related infection (vascular graft)	*Staphylococcus epidermidis*	Unknown	-	4: 1000, 500, 500, 500 On day 0, 7, 15, 22	N/A (first-line)	Success
8	Aortic graft infection, Gram-positive bacteraemia	*Staphylococcus haemolyticus*	Fosfomycin, Flucloxacillin	-	1: 1500	Insufficient therapeutic effect of prior therapy	Success
9	Prosthetic joint infection, Gram-positive bacteraemia	*Staphylococcus lugdunensis*, *Escherichia coli*	Ciprofloxacin, Rifampicin, Ampicillin/Sulbactam, Linezolid, Fosfomycin, Daptomycin	Amoxicillin	4: 1500, 1500, 1500, 1500 On day 0, 14, 33, 54	Adverse reaction to prior therapy	Success

ABSSSI, acute skin and skin structure infections.

**Table 3 antibiotics-11-00563-t003:** Overview of adverse events (all patients).

Adverse Event	Seriousness (Criterion)	Relatedness to Dalbavancin	Severity	Outcome	All Patients (*N* = 16) *n* (%)—Events
Multiple organ dysfunction syndrome	Serious (Fatal/Life-threatening/Hospitalization)	Not related	Not severe	Fatal	1 (11.1%)—1
*Candida* infection	Serious (Hospitalization)	Not related	Not severe	Recovered/Resolved	1 (11.1%)—1

**Table 4 antibiotics-11-00563-t004:** Previous studies on dalbavancin.

Reference	Type of Study	Patients: *N*, Age, Sex, Charlson Comorbidity Index (CCI)	Primary Disease	Dalbavancin Regimen	Outcome
Matt et al., 2021: [20]	Observational national (French) cohort study and literature review	*N* = 17; median age 69.0 years (IQR 62.0–75.0); sex ratio male/female 1.43; CCI not analyzed	PJI	Preferential administration scheme of dalbavancin: 1500 mg at day 0 and 1500 mg at day 7 (47.1%)	Clinical cure: 8/17 (47%) patients; clinical cure in PJI patients in the literature: 73.1%
Poliseno et al., 2021: [9]	Retrospective cohort study (patients of the University Hospital Policlinico of Bari, Italy)	*N* = 50; median age 61 years (IQR 51–75); sex 68% male; median CCI 3 (IQR 1–4)	Diverse Gram-positive bacterial infections (including 12 patients with ABSSSI and 8 patients with complicated ABSSSI)	Median number of dalbavancin 1500 mg doses administered per patient was 1 (IQR 1–3), but significant inter-subject differences were observed	Clinical success: 49/50 (98%) patients
Veve et al., 2020: [21]	Retrospective cohort study	*N* = 70 (dalbavancin arm); median age 47 years (range 36–55); sex 60% male; CCI not analyzed	Osteoarticular infection, infective endocarditis, other bloodstream infection	The most frequently used dalbavancin doses were 1500 mg for two doses 1 week apart (34%), 1500 mg for one dose (26%), 1500 mg for two doses 2 weeks apart (21%), and other doses less than 1500 mg (19%)	Dalbavancin use associated with lower 90-day infection-related readmission (IRR), a shorter hospital length of stay prior to therapy, and longer time-to-IRR compared with standard of care
Bai et al., 2020: [16]	Italian observational multicentre study (DALBITA study)	*N* = 206; median age 62 years (IQR 50–76); sex 50% male; median CCI 3 (IQR 1–5)	ABSSSI (60%) and ‘other sites’ infections’ (40%)	‘Standard dosage’: 1500 mg single-dose (60.2%)); the maximum number of weekly repetitions was 7	Clinical cure: 170/206 (83%) patients
Buzón Martín et al., 2019: [22]	Retrospective cohort study	*N* = 16; median age 76 years (IQR 70.25–82.25); sex 56.2% male; median age-adjusted CCI 3 (IQR 3–5)	PJI	56% of patients received 1500 mg of dalbavancin as loading dose, followed by 500 mg at day seven, and then 500 mg every two weeks (i.e., *l**ow-dose* *bi-weekly dalbavancin)*	Infection resolved: 12/16 (75%) patients; (treatment failure: two patients; one patient died due to unrelated causes; one patient haematogenously spread knee infection secondary to prosthetic aortic arch endocarditis)
Tubudic et al., 2019: [23]	Case series (observed at the University Hospital of Vienna)	*N* = 72; median age 56.5 years (range 18–92); sex 53% male; CCI not analyzed	SSTI (36%), osteomyelitis (28%), spondylodiscitis (19%), acute septic arthritis (6%), PJI (11%)	Most common regimen used: initial dose 1500 mg and 1000 mg every 14 days (71%)	Clinical cure: 46/72 (64%) patients; (among 26 patients who received additional antibiotic therapy other than dalbavancin, 15 patients [21%] showed no clinical improvement under dalbavancin therapy)
Dinh et al., 2019: [12]	National (French) retrospective study of all adult patients who received at least one dose of dalbavancin	*N* = 75; mean age 63.1 years (SD 17.0); sex ratio male/female 2.26; CCI not analyzed	Main sites of infection: bone and joint infection, endocarditis, SSTI; concomitant bacteraemia in 51%	Main dalbavancin treatment regimens: two 1500 mg injections with a 7-day interval (48%) or a 14-day interval (11%), and a single 1500 mg injection (13%)	Clinical cure: 54/68 (79%) patients: failure: 14/68 patients
Wunsch et al., 2019: [13]	Multicentre, retrospective study in Austria	*N* = 101; median age 65 years (range 11–93); sex 56.4% male; CCI not analyzed	PJI (31%), osteomyelitis (29%), endocarditis (25%), acute bacterial skin and soft tissue infections (11%), catheter-related bloodstream infections (3%)	Mostly a single 1000 mg dose at day 0 followed by 500 mg weekly	Clinical success rate at day 90: 84/94 (89%) patients
Almangour et al., 2019: [24]	Multicentre retrospective review	*N* = 31; mean age 50 years (SD 14); sex 74% male; CCI not analyzed	Osteomyelitis	Number of dalbavancin doses varied from a single dose to 14 doses (median = 3; IQR = 3). Doses ranged from 500 to 1500 mg per dose	Clinical success: 28/31 (90%) patients
Bouza et al., 2018: [11]	Observational retrospective study (in 29 hospitals from 14 urban centers in Spain)	*N* = 69; median age 63.5 years (range 15–90; IQR 49.3–72.0); sex 58.0% male; median CCI 3 (IQR 1–5)	Most common infections: PJI (29%), ABSSSI (22%), osteomyelitis (17%), catheter-related bacteraemia (12%), endocarditis (10%); concomitant bacteraemia in 26%	Mostly a dose of 1000 mg at day 0 followed by 500 mg weekly to cover 14 to 42 days	Clinical success: 58/69 (84%) patients; clinical failure: 11/69 (16%) patients

ABSSSI, acute skin and skin structure infections; CCI, Charlson Comorbidity Index; IQR, interquartile range; IRR, infection-related readmission; PJI, prosthetic joint infection; SSTI, skin and soft tissue infection; SD, standard deviation. Note: This list is a selection of real-world observational studies without a specific focus on pharmacokinetics/pharmacodynamics or health economics.

## Data Availability

All relevant data are within the manuscript. Furthermore, data that support the findings of this study are available from the corresponding author upon reasonable request.

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
