# Peer review of "A Multicentre, Prospective, and Retrospective Registry to Characterize the Use, Effectiveness, and Safety of Dalbavancin in German Clinical Practice"

_antibiotics, 2022, doi:10.3390/antibiotics11050563_

Round 1

Reviewer 1 Report

In this study, the authors report the data regarding the effectiveness and safety of the lipoglycopeptide dalbavancin in German clinical practice. The data described concern 9 patients, recruited in one year. Initially, the study was to be much larger including 150 patients recruited in two years. Hence, my biggest concern is the statistical validity of the reported data. However, the authors cope very well with this weakness, highlighting how this is only a preliminary study and comparing the results with others reported in the literature. Overall, the results are clear, the methods are accurately described, and the paper sounds good.

There are some aspects that can be improved such as:

1. Since this study includes a limited amount of data and is, therefore, preliminary, this aspect should be made explicit in the title and abstract.

2. The document seems to focus on the "Dalbavancin Utilization Registry in Germany". Please, add details on this Registry. What implications will this register have in the future?

3. Other statistical methods could be used to improve the study. I believe that a deeper analysis of the data at a statistical level could strengthen the study. For example: what is the probability that positive results are "false positives"?

4. Are all the dalbavancin studies shown in table 4? What criteria were used for the selection of the studies reported in this table?

Reviewer 2 Report

Dear Authors

Good Day. 

Greetings. 

How do calculate sample size? 

What were the sampling methods? 

There is no control. Therefore no possibility of comparison. Please justify? 

So far my understanding is it will be difficult to generalize data for the whole community. Please mention with a separate heading study limitation.

Kindly remove the brand name of the antibiotic. 

Please add a future implication of this paper 

Good Luck with your paper. 
